# High-Throughput Screening of FDA-Approved Drug Library Reveals Ixazomib Is a Broad-Spectrum Antiviral Agent against Arboviruses

**DOI:** 10.3390/v14071381

**Published:** 2022-06-24

**Authors:** Cuiling Ding, Wanda Tang, Binghui Xia, Haoran Peng, Yan Liu, Jiaqi Wang, Xu Zheng, Yangang Liu, Lanjuan Zhao, Yanhua He, Zhongtian Qi, Hao Ren, Hailin Tang, Ping Zhao

**Affiliations:** Department of Microbiology, Navy Medical University, Shanghai 200433, China; cuilingding@163.com (C.D.); twd4250811@163.com (W.T.); xbhnjucpu@163.com (B.X.); phran@126.com (H.P.); 15801952518@163.com (Y.L.); wjq199207@163.com (J.W.); zx_tjnxy@126.com (X.Z.); lyg@smmu.edu.cn (Y.L.); ljzhao13@163.com (L.Z.); hyh2624@163.com (Y.H.); qizt@smmu.edu.cn (Z.Q.); hren2020@163.com (H.R.)

**Keywords:** arboviruses, ixazomib, broad-spectrum antivirals screening, FDA-approved drugs

## Abstract

The emergence of significant arboviruses and their spillover transmission to humans represent a major threat to global public health. No approved drugs are available for the treatment of significant arboviruses in circulation today. The repurposing of clinically approved drugs is one of the most rapid and promising strategies in the identification of effective treatments for diseases caused by arboviruses. Here, we screened small-molecule compounds with anti-tick-borne encephalitis virus, West Nile virus, yellow fever virus and chikungunya virus activity from 2580 FDA-approved drugs. In total, 60 compounds showed antiviral efficacy against all four of the arboviruses in Huh7 cells. Among these compounds, ixazomib and ixazomib citrate (inhibitors of 20S proteasome β5) exerted antiviral effects at a low-micromolar concentration. The time-of-drug-addition assay suggested that ixazomib and ixazomib citrate disturbed multiple processes in viruses’ life cycles. Furthermore, ixazomib and ixazomib citrate potently inhibited chikungunya virus replication and relieved virus-induced footpad swelling in a mouse model. These results offer critical information which supports the role of ixazomib as a broad-spectrum agent against arboviruses.

## 1. Introduction

In recent years, the emergence of arboviruses and their spillover transmission to humans have been causing major health and economic problems worldwide. In particular, re-emerging flaviviruses, such as tick-borne encephalitis virus (TBEV), West Nile virus (WNV) and yellow fever virus (YFV), and alphaviruses, such as chikungunya virus (CHIKV), are arboviruses that are causative of severe and life-threatening diseases. Flaviviruses and alphaviruses are small, enveloped, single-stranded, positive-sense RNA viruses transmitted primarily by *Aedes* spp. mosquitoes [1]. As part of their viral replication cycle, TBEV, WNV, YFV and CHIKV enter host cells via viral glycoprotein receptor-mediated endocytosis to infect cells and then synthesize viral proteins and replicate their genome [2]. No approved drugs or vaccines are available for the treatment of many of the most important arboviruses in circulation today; therefore, efforts to prevent these diseases are urgently needed and must be addressed.

As de novo antiviral drug development requires long R&D ranges and large investments, drug repurposing is one of the most rapid and promising strategies for the identification of effective treatments for diseases caused by pathogens [3]. Additionally, the repurposing of clinically approved drugs for application in the treatment of an emerging or re-emerging viral infection is a method that is widely used [4,5,6,7]. Here, we carried out high-throughput repurposing screening from a FDA-approved drug library containing 2580 small-molecule compounds to identify broad-spectrum antivirals for TBEV, WNV, YFV and CHIKV. In total, 60 molecules that showed antiviral effects against these four arboviruses were identified. Ixazomib and ixazomib citrate (inhibitors of 20S proteasome β5) exerted high antiviral potency in vitro, showing low-micromolar antiviral effects. Additionally, the role of ixazomib and ixazomib citrate in the life cycle of viruses was investigated further. The anti-CHIKV activities of ixazomib and ixazomib citrate in vivo were evaluated using a mouse model. To our knowledge, this study is the first to describe the antiviral efficacy of ixazomib against emerging flaviviruses and alphaviruses, and it demonstrates that ixazomib is a promising broad-spectrum antiviral candidate.

## 2. Materials and Methods

### 2.1. Cell Culture and Viruses

Human hepatoma Huh7 cells (SCSP-526, Chinese Academy of Sciences, Shanghai, China) were cultured in Dulbecco’s modified Eagle’s medium (Life Technologies, Carlsbad, CA, USA), supplemented with 10% fetal bovine serum (Gibco, Carlsbad, CA, USA), 1% penicillin/streptomycin (Gibco, Carlsbad, CA, USA), 1% L-glutamine (Gibco, Carlsbad, CA, USA) and 1% non-essential amino acids (Gibco, Carlsbad, CA, USA) at 37 °C in a humidified atmosphere with 5% CO2. A tick-borne encephalitis virus (TBEV)-infectious clone of the Zmeinogorsk-5 strain (KY069125), a WNV-infectious clone of the NY2000 strain (AF404756) and a chikungunya virus (CHIKV)-infectious clone of the LR2006 OPY1 strain were synthesized in this laboratory. Yellow fever virus (YFV0 strain) was isolated from the serum of a confirmed YF patient (FJYF03/2016, GenBank: KY587416.1).

### 2.2. Compound Library

A Selleck library of FDA-approved drugs (L1300-Z349373), consisting of 2580 compounds, was purchased from Selleck Chemicals (Houston, TX, USA). All of the compounds were dissolved in dimethyl sulfoxide (DMSO) as 10 mM stock solutions and stored at −80 °C.

### 2.3. Drug Screening

All of the compounds were diluted in culture media to a final concentration of 5 μM during screening. Briefly, a total of 1 × 10^4^ Huh7 cells were seeded in 96-well plates. Then, 12 h later, the cells were treated with compounds at 37 °C and 5% CO_2_ prior to infection with a virus at a MOI of 0.1. All of the treatments were conducted in triplicate for each compound. DMSO was used as a control. Next, 24 h post infection, the cells were fixed with methyl alcohol, and then, immunofluorescence imaging was performed to determine the infection rate of the viruses. Rabbit polyclonal antibodies against the virus were prepared by immunizing rabbits with formaldehyde-inactivated viruses; the Alexa Fluor 488-conjugated goat-anti-rabbit IgG (Thermo Fisher Scientific, Waltham, MA, USA); DAPI (1:10,000 dilution, Sigma-Aldrich, Darmstadt, Germany). Images were acquired using Cytation 5 (BioTek, El Segundo, CA, USA). The numbers of infected cells and the total cells in each well were counted using Gen5 3.10. Then, the infection rate of each compound-treated group was calculated in Prism.

### 2.4. Dose–Response Studies and EC50 Calculation

The antiviral activity of the selected compounds was further validated by carrying out dose–response studies. In total, 1 × 10^4^ Huh7 cells were seeded in 96-well plates. Then, 12 h later, the cells were treated with eight 2-fold serial dilutions of compounds. The final concentration of the compounds ranged from 20 μM to 0.15625 μM. DMSO was used as a control. Viruses’ working stocks were added to the treated Huh7 cells at a MOI of 0.1. Next, 24 h after the virus incubation, the infection rate of the viruses was determined using immunofluorescence imaging, as described above. The concentration of 50% of the maximal effect (EC50) of each compound was calculated in Prism. In addition, parallel plates that were not infected were analyzed to monitor the cytotoxicity of each compound. The cells were maintained at 37 °C and at 5% CO_2_ for 48 h before a CCK-8 assay was performed, following the manufacturer’s instructions (Beyotime, Shanghai, China). The A450 was read using Synergy 2 (BioTek, El Segundo, CA, USA). The CC50 of each compound was calculated in Prism.

### 2.5. Time-of-Drug-Addition Assay

To carry out the time-of-drug-addition assay, 1 × 10^4^ Huh-7 cells were seeded in 96-well plates in advance. Then, 12 h later, each compound (5 μM) was added 2 h before virus infection, simultaneously with the virus addition and 2, 4, 6, 8, 10 and 12 h after virus infection, and the compound action period was 2 h (the action period of adding compounds after 12 h of virus infection was 12 h). The virus incubation period was 2 h. DMSO was used as a control. After the drug treatment, the medium was replaced with fresh DMEM with 10% fetal bovine serum for 24 h. After fixation with methanol solution, the infection rates were detected using immunofluorescence imaging, as described above.

### 2.6. Mouse Experiment

Four-week-old female C57BL/6 mice were used for the in vivo evaluation of the anti-CHIKV activity of the compounds. All of the procedures that involved animals were reviewed and approved by the Institutional Committee for Animal Care and Biosafety of Navy Medical University. All of the experiments complied with the relevant ethical regulations. All of the mouse infection studies were performed in a laboratory with an animal biosafety level of 3. For the animal experiments, the C57BL/6 mice were randomly distributed into 4 groups. Each group consisted of nine mice. Three groups of mice were inoculated subcutaneously in the left rear foot with 10^6^ focus-forming units (FFUs) of the CHIKV LR2006 OPY1 strain, as described previously [8]. One group was used as a normal control, which did not undergo virus infection. Starting on day 1 after CHIKV inoculation, mice in groups 1–3 received different compounds at appropriate doses: vehicle, ixazomib (7 mg/kg) or ixazomib citrate (11 mg/kg) [9,10], via oral administration, once daily for 4 consecutive days. Three mice were monitored daily for left rear footpad swelling with digital calipers for 9 days. Three mice per group were sacrificed at 3 days post infection (dpi) and 5 dpi; their hamstring muscle tissues from the left rear foot were collected for virus load evaluation, and the sera were collected for viremia evaluation, as described previously [11]. The viral load and inflammation in the muscle tissues were determined using quantitative real-time RT-PCR (qRT-PCR) methods. The qRT-PCR of virus RNA was performed using SYBR Premix Ex TaqTMII (Tli RNaseH Plus) (TaKaRa, Kusatsu, Japan). The expression of target genes was normalized to GAPDH mRNA levels in the same samples. The primer sequences were as follows: CHIKV (genome location: 3171–3276 bases), 5′-GGCAGTGGTCCCAGATAATTCAAG-3′ and 5′-ACTGTCTAGATCCACCCCATACATG-3′; GAPDH, 5′-TGGGCTACACTGAGCACCAG-3′ and 5′-AAGTGGTCGTTGAGGGCAAT-3′; monocyte chemoattractant protein 1 (MCP-1), 5′-CAGCCAGATGCAGTTAACGC-3′ and 5′-CAGACCTCTCTCTTGAGCTTGG-3′; tumor necrosis factor alpha (TNF-α), 5′-AATTCGAGTGACAAGCCTGTAGC-3′ and 5′-AGTAGACAAGGTACAACCCATCG-3′.

### 2.7. Statistical Analysis

Bar and line graphs showing the means ± SEMs of at least three independent experiments were plotted. Statistical analyses were performed using Prism 5 (GraphPad Software). Data were analyzed using either the Student’s *t*-test or one-way ANOVA. *p*-values are indicated by *, *p* < 0.05, **, *p* < 0.01 and ***, *p* < 0.001.

## 3. Results

### 3.1. High-Throughput Screening of Anti-TBEV, WNV, YFV and CHIKV Compounds

A high-throughput screening assay was performed to screen antiviral candidates against TBEV, WNV, YFV and CHIKV from 2580 FDA-approved small-molecule drugs. Huh7 cells and TBEV, WNV, YFV and CHIKV viruses were used for this screening. Figure 1a shows the procedure used in the screening assay. In total, 60 compounds showed potent antiviral activity against TBEV, WNV, YFV and CHIKV, meaning they could inhibit the infection of the four viruses by more than 90% in Huh7 cells at a final concentration of 5 μM. To validate the antiviral activity of these 60 compounds, monkey kidney epithelial cells, Vero cells, were utilized. Thrillingly, all of the compounds exerted inhibitory effects against the four viruses by more than 85% in Vero cells at a final concentration of 5 μM, which suggested that the antiviral activities of these compounds were not cell-dependent. Ivacaftor (an enhancer of the cystic fibrosis transmembrane conductance regulator) was identified to show antiviral activity for the first time. Additionally, two compounds, telotristat etiprate and sonodegib, which have also been reported to inhibit SARS-CoV-2 in screening studies, were identified to inhibit the infection of TBEV, WNV, YFV and CHIKV [12,13]. The enrichment of the known targets of these 60 drugs was analyzed and classified (Appendix A). Six main targets were classified: receptor tyrosine kinase (RTK), 20S proteasome β5, ion channel, DNA synthesis, antibacteria or protozoa drugs and natural products (Figure 1b). Numerous studies report RTK inhibitors, ion channel inhibitors, DNA synthesis inhibitors, antibacteria or protozoa drugs and natural products to be arbovirus antivirals [14,15,16,17]. However, little work has been conducted regarding 20S proteasome β5 inhibitors as direct-acting antivirals. In this study, the antiviral efficacy of three 20S proteasome β5 inhibitors, ixazomib, ixazomib citrate and carfilzomib, was further evaluated.

### 3.2. Dose–Response Analysis

The 50% effective concentration (EC50) and 50% cytotoxic concentration (CC50) values of three 20S proteasome β5 inhibitors against TBEV, WNV, YFV and CHIKV were further evaluated in Huh7 cells. The working concentration of the drugs started at 20 μM, and eight 2-fold serial dilutions were performed (20–0.15625 μM). The CC50 of ixazomib and ixazomib citrate in Huh7 cells was shown to be 12.62 μM and 10.13 μM, respectively (Figure 2a,b). However, the cytotoxicity of carfilzomib was overt, being 0.15625 μM (Figure 2c). Thus, carfilzomib was excluded in the following study. Remarkably, the EC50 values of ixazomib and ixazomib citrate against these four viruses were found to lie in the submicromolar range (Figure 3). Additionally, the selective index (SI) of ixazomib against TBEV, WNV, YFV and CHIKV was 324.59, 125.198, 77.41 and 68.36, respectively. Ixazomib citrate also displayed antiviral activity that inhibited TBEV, WNV, YFV and CHIKV with the SI of 410.12, 98.16, 55.17 and 55.05, which is comparable with ixazomib.

### 3.3. Effects of Ixazomib on the Virus Life Cycle

To investigate which stages of the TBEV, WNV, YFV and CHIKV life cycles were interrupted by ixazomib, a time-of-drug-addition assay was performed by treating virus-infected Huh7 cells with ixazomib or ixazomib citrate at various virus-specific time points, followed by immunofluorescence imaging 24 h post infection (hpi) (Figure 4a). Ixazomib inhibited TBEV infection in multiple periods. The prominent suppressive activity of ixazomib against TBEV occurred when it was maintained at 0–2 hpi, the stage of cell entry after attachment, which revealed that ixazomib inhibited mostly TBEV in the entry stage (Figure 4b). WNV and YFV infection was disrupted by ixazomib when it was maintained at 0–2 hpi and 2–4 hpi, suggesting ixazomib was involved in the viral entry and replication processes of both viruses (Figure 4c,d). Intriguingly, ixazomib significantly suppressed TBEV, WNV and YFV at 0–2 hpi, while ixazomib showed no inhibitory effect for CHIKV when it was maintained between 0 and 2 hpi, but it inhibited CHIKV infection when it was maintained at pre 2–0 hpi and 2–10 hpi (Figure 4e). The best suppressive activity was observed at 4–8 hpi, which demonstrated that ixazomib mostly interfered with the replication and package of CHIKV. At present, there are no relevant literature reports regarding the role of the 20S proteasome β5 in the infection process of TBEV, WNV, YFV and CHIKV. Whether ixazomib inhibited virus infection by suppressing the activity of 20S proteasome β5 remains unclear. Additionally, the mechanisms of ixazomib inhibiting TBEV, WNV, YFV and CHIKV need to be investigated further.

### 3.4. Ixazomib Alleviated the Footpad Swelling Caused by CHIKV Infection

To evaluate the antiviral effect of ixazomib in vivo further, a CHIKV infection mouse model was utilized. Four-week-old female C57BL/6 mice were divided into four groups: three groups received one of the following treatments beginning on day 1 after infection: vehicle, ixazomib (7 mg/kg) or ixazomib citrate (11 mg/kg), via oral administration at suitable doses; one group was used as a normal control group which did not undergo CHIKV infection. Animals were monitored for footpad swelling in the left rear foot, and viral loads in the left rear hamstring muscle were monitored at different times after infection and treatment. CHIKV-infected mice showed one peak of clinically apparent footpad swelling 6 days after infection. Compared with the vehicle-treated group, ixazomib and ixazomib citrate significantly alleviated the footpad swelling caused by CHIKV infection (Figure 5b). Additionally, the virus loads in the left rear hamstring muscles of mice with ixazomib and ixazomib citrate treatments were distinctly reduced compared with those of the vehicle-treated group when assayed on 3 dpi and 5 dpi (Figure 5c). The results demonstrated that ixazomib can effectively alleviate footpad swelling and suppress viral replication in mouse muscles. Concurrently, compared with the vehicle group, ixazomib or ixazomib citrate treatment decreased the viremia at 3 dpi (Figure 5d), while no CHIKV infection was detected at 5 dpi in all three groups (data not shown). In addition, the mRNA expression levels of monocyte chemoattractant protein 1 (MCP-1) and tumor necrosis factor alpha (TNF-α) in the left rear hamstring muscle tissues were determined. At 3 dpi and 5 dpi, the MCP-1 and TNF-α mRNA expression levels were remarkably diminished in the mice treated with ixazomib and ixazomib citrate (Figure 5e,f). Taken together, ixazomib and ixazomib citrate inhibited CHIKV infection in vivo and attenuated inflammation caused by CHIKV.

## 4. Discussion

Arboviral infections have had profound effects on public health in many parts of the world. TBEV, WNV, YFV and CHIKV pose continuous health threats. In the face of the rapid rise in the number of emerging and reemerging infectious diseases and the lack of vaccines and drugs, there is an urgent need for therapeutic approaches that can target large classes of viruses using single drugs. Compared with novel drug development, drug repurposing has several advantages: it is faster, has good safety levels, causes few side effects and is inexpensive; these advantages mean that it is a plausible way to identify treatment candidates for emerging and re-emerging infectious diseases.

In this study, a high-throughput screening assay was used to identify anti-TBEV, WNV, YFV and CHIKV candidates from 2580 FDA-approved small-molecule drugs, and 60 compounds were found to inhibit the infection of the four above mentioned arboviruses by more than 90% at a concentration of 5 μM in Huh7 cells. Although many of these compounds overlapped with several recent drug repurposing screening efforts for different viruses [18,19], for the first time, ivacaftor (an enhancer of the cystic fibrosis transmembrane conductance regulator) showed antiviral activity. Additionally, telotristat etiprate (a tryptophan hydroxylase inhibitor) and sonodegib (a smoothened antagonist) were identified to inhibit the infection of TBEV, WNV, YFV or CHIKV for the first time. The targets of these 60 compounds were classified, and the pathways that these targets are involved in were analyzed. Notably, VEGFR is a pivotal target of a variety of viruses [20], which is the largest target group that was identified in this study.

The ubiquitin–proteasome system (UPS) is critical in the maintenance of cellular homeostasis. As such, viruses exploit the UPS to manipulate host cells and then benefit their own infectious cycles. Proteasome activity is required in multiple stages of many different viruses’ infections [21,22,23]. The UPS is reported to be involved in the life cycle process in different viruses. Proteasome-mediated stages of the initiation of the infection include endosomal escape, nucleocapsid trafficking and the uncoating of the viral genome [21]. Japanese encephalitis virus (JEV) is a member of the flavivirus genus. The UPS was shown to be involved in the endosomal escape stage of JEV entry [24]. The transport of simian virus 49 (SV49) [25,26] and BK polyomavirus (BKV) [27] capsids from ER to the cytosol was shown to be dependent on UPS functions. Dengue virus (DENV) is another member of the flavivirus genus, and its uncoating stage (viral nucleic acid typically separates from viral core proteins to initiate viral replication) was reported to be likely affected by the UPS [28]. In addition, many studies focus on viral genome replication that is dependent on UPS function [29,30,31]. The 26S proteasome is at the heart of the ubiquitin–proteasome pathway and consists of a catalytic 20S proteasome and an ATP-dependent 19S cap [32]. The 20S proteasome contains three proteolytic subunits with distinct caspase-like (β1), trypsin-like (β2) and chymotrypsin-like (β5) activities [33,34]. However, the role of the β5 subunit of the 20S proteasome in the life cycle of viruses is complex and unclear.

Ixazomib citrate is the first orally bioavailable inhibitor of the 20S proteasome to be evaluated for the treatment of multiple myeloma [35]. It is more selective for the chymotrypsin-like (β5) protease, causing less inhibitory activity against other active subunits [36]. Ixazomib citrate undergoes rapid hydrolysis to biologically active boronic acid, ixazomib, under physiological conditions [37]. Ixazomib is reported to reactivate latent HIV and predispose reactivated cells to cell death [38]. Additionally, ixazomib was evaluated for use in the treatment of EBV-associated B-cell neoplasms to target NF-κB signaling [39]. It was also reported to be involved in the TRAIL/death receptor signaling and β-catenin/TCF signaling pathways [40,41]. In our study, both ixazomib and ixazomib citrate showed excellent antiviral efficacy toward TBEV, WNV, YFV and CHIKV, with submicromolar EC50 data. Furthermore, ixazomib was involved in multiple stages of the TBEV and CHIKV life cycles and the early stages of the WNV and YFV life cycles. Of note, ixazomib inhibited TBEV infection with the best suppressive effect when maintained at 0–2 hpi after virus infection, the important entry stage after attachment, which demonstrated that ixazomib mostly inhibited TBEV in the entry stage. It is very interesting that ixazomib did not inhibit CHIKV infection when maintained at 0–2 hpi, but it did at 2–8 hpi. Moreover, the antiviral effect was most apparent when it was maintained between 4 and 8 hpi, which suggested that ixazomib may disturb the package of CHIKV particles. This study revealed that ixazomib performs different roles in different virus infection processes and has multiple roles in one virus life cycle. Additionally, it is worth considering that ixazomib may inhibit TBEV, WNV, YFV and CHIKV through UPS or through other targets, which requires further study.

In order to evaluate the antiviral activity of ixazomib and ixazomib citrate in vivo, a CHIKV arthritis mouse model was utilized. C57BL/6 mice are susceptible to CHIKV infection, which has been widely used in studies concerning the pathogenicity of CHIKV [42,43]. The results showed that both ixazomib and ixazomib citrate treatments effectively ameliorated footpad swelling and reduced muscle viral burdens at 3 dpi and 5 dpi. The results confirmed the antiviral activities of ixazomib and ixazomib citrate in vivo. Future research will aim to evaluate the antiviral effects of ixazomib against different arboviruses in vivo. As a FDA-approved drug for the treatment of multiple myeloma, the toxicity of ixazomib was systematically evaluated [44]. While, as an anticancer drug, the toxicity is still a concern of ixazomib to be used for anti-virus infection in clinical settings, finding a safe therapeutic window may be feasible [45]. Before ixazomib is applied as an antiviral agent in clinical settings, much more work is still required.

In summary, we discovered a list of clinically approved drugs capable of inhibiting TBEV, WNV, YFV, and CHIKV in vitro in Huh7 cell lines. Moreover, we showed that ixazomib is involved in multiple stages of virus infection processes and performs different roles in different virus infection processes. Furthermore, the anti-CHIKV efficacy of ixazomib was validated in vivo using a mouse model. Our results showed that ixazomib may be a promising broad-spectrum antiviral agent for use against arbovirus infections.

## Figures and Tables

**Figure 1 viruses-14-01381-f001:**
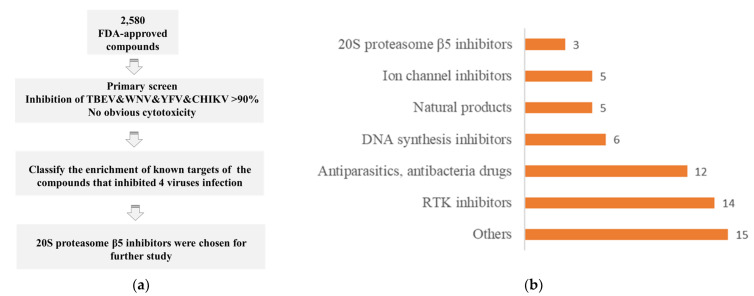
High-throughput screening for anti-TBEV, WNV, YFV and CHIKV compounds. (**a**) Schematic diagram of high-throughput drug screening for anti-TBEV, WNV, YFV and CHIKV compounds and further study. (**b**) Molecular targets of compounds with anti-TBEV, WNV, YFV and CHIKV activity.

**Figure 2 viruses-14-01381-f002:**
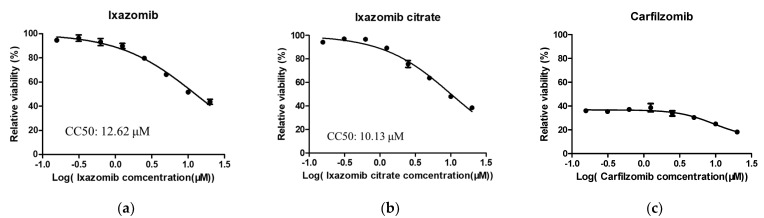
(**a**) Ixazomib; (**b**) Ixazomib citrate and (**c**) Carfilzomib: Cell viability detection of three 20S proteasome β5 inhibitors in Huh7 cells. Huh7 cells were treated with eight 2-fold serial dilutions (20–0.15625 μM) of three 20S proteasome β5 inhibitors. The cells were maintained at 37 °C and at 5% CO_2_ for 48 h before a CCK-8 assay was performed. Data are normalized to the average of DMSO-treated wells and represent mean ± SEM for *n* = 3 independent experiments.

**Figure 3 viruses-14-01381-f003:**
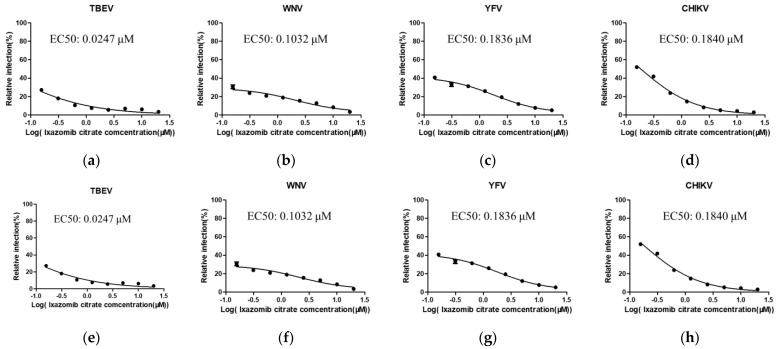
Dose–response relationship of ixazomib (**a**–**d**) and ixazomib citrate (**e**–**h**) regarding anti-TBEV, WNV, YFV and CHIKV activity in Huh7 cells. Huh7 cells were treated with eight 2-fold serial dilutions (20–0.15625 μM) of ixazomib or ixazomib citrate and then infected with indicated virus at a MOI of 0.1. Then, 24 h post infection, the cells were fixed and visualized using immunofluorescence imaging. For each condition, the percentage of infection was calculated as the ratio of the number of infected cells stained for virus to the number of cells stained with DAPI. Data are normalized to the average of DMSO–treated wells and represent mean ± SEM for *n* = 3 independent experiments.

**Figure 4 viruses-14-01381-f004:**
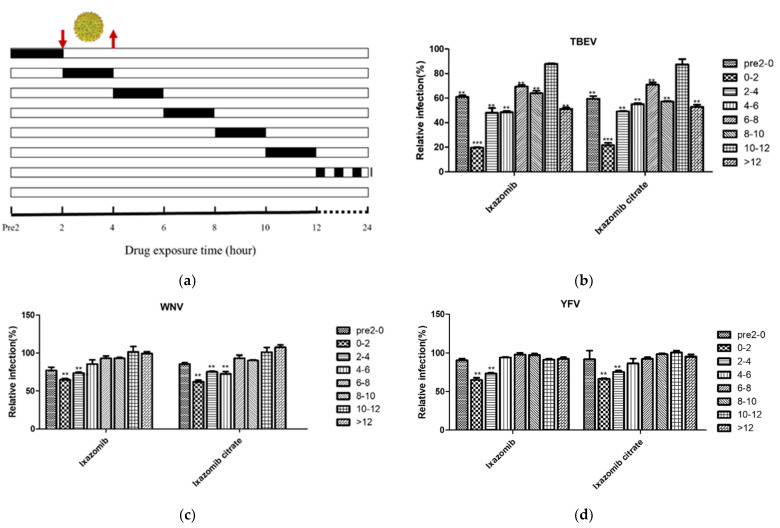
Effect of compounds on TBEV (**b**), WNV (**c**), YFV (**d**) or CHIKV (**e**) life cycles, determined using a time–of–drug–addition assay. (**a**) Schematic diagram of the time–of–drug–addition assay. Huh7 cells were infected with virus (MOI = 0.1) and treated with 5 μM ixazomib or ixazomib citrate during different periods of indicated viral infection. At 24 h post infection, cells were analyzed using immunofluorescence imaging. Data are normalized to the average of DMSO–treated wells and represent mean ± SEM for *n* = 3 independent experiments. *p*-values are indicated by **, *p* < 0.01 and ***, *p* < 0.001.

**Figure 5 viruses-14-01381-f005:**
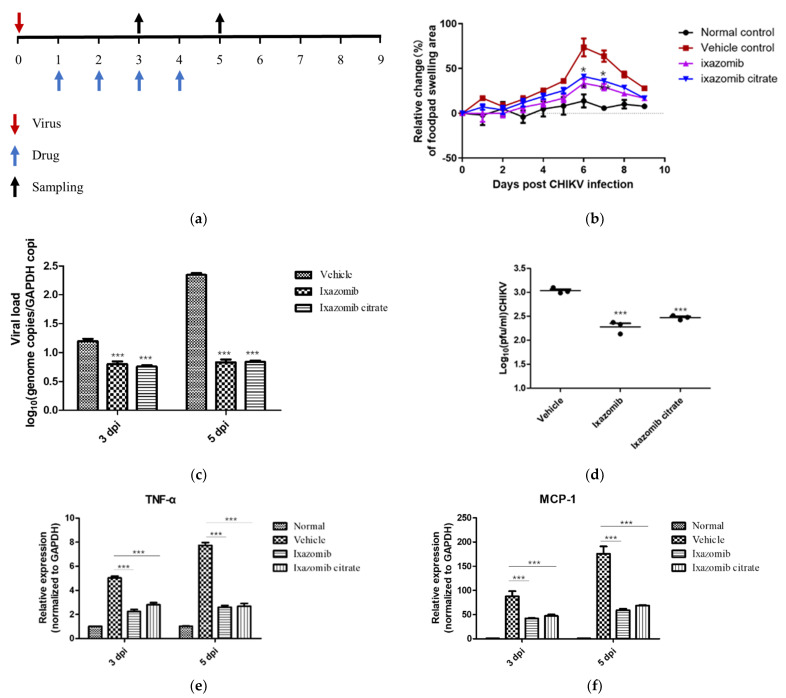
Antiviral activity of ixazomib and ixazomib citrate in CHIKV-infected C57BL/6 mice. (**a**) Schedule of drug treatments and CHIKV infection in mice. Mice were inoculated subcutaneously in the left rear feet with 10^6^ focus-forming units (FFUs) of CHIKV LR2006 OPY1 strain. (**b**) Four groups of mice were monitored daily for left rear footpad swelling with digital calipers for 9 days. (**c**) Viral loads in the muscle tissues at 3 dpi were measured by determining the genome copies/GAPDH copies using qRT–PCR methods. (**d**) Viremia were measured at 3 dpi. Relative mRNA expression of TNF–α (**e**) and MCP–1 (**f**) in the muscle tissues of the indicated groups (*n* = 3), as detected in the mice muscle tissues at 3 dpi and 5 dpi. Data were shown as mean ± SEM. * *p* < 0.05, ** *p* < 0.01, *** *p* < 0.001 compared to vehicle group.

## Data Availability

The data that support the findings of this study are available from the corresponding author upon reasonable request.

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
