# Peer review of "High-Throughput Screening of FDA-Approved Drug Library Reveals Ixazomib Is a Broad-Spectrum Antiviral Agent against Arboviruses"

_viruses, 2022, doi:10.3390/v14071381_

Round 1
Reviewer 1 Report
The authors described the screening of the small-molecule compounds against TBEV, WNV, YFV and CHIKV replication from 2,580 FDA-approved drugs, and found that ixazomib and ixazomib citrate exerted antiviral activity against four viruses mentioned above. Additionally, they found that both compounds could inhibit viral infection in mice. Overall, the manuscript is well preparedexcept that some improvements are needed before consideration for publication.
1. In animal experiments of drug inhibition of chikungunya virus, only the swelling of footpad and the left rear hamstring muscle viral burden were observed. It is suggested to detect more indicators, such as viremia, Histological analysis of joint footpad samples, etc.
2. Whether the drug inhibits the virus due to its the 20S proteasome β5 inhibitors status needs to be verified experimentally.
Author Response
Response to Reviewer 1 comments:
Point 1: In animal experiments of drug inhibition of chikungunya virus, only the swelling of footpad and the left rear hamstring muscle viral burden were observed. It is suggested to detect more indicators, such as viremia, Histological analysis of joint footpad samples, etc.
Response 1: We are grateful for the valuable suggestion. In order to further evaluate the inhibitory effect of ixazomib and ixazomib citrate against CHIKV in mice model, the viremia and the expression of 2 inflammatory factors, MCP-1 and TNF-α, in hamstring muscle tissues were tested. The results were shown in Figure 5d, 5e and 5f in the revised manuscript. Compared with vehicle-treatment group, ixazomib- or ixazomib citrate-treatment decreased the viremia at 3 dpi (Figure 5d). While, no viremia was detected at 5 dpi in all three groups (data not shown). And, the mRNA expression levels of MCP-1 and TNF-α were remarkably diminished in the mice treated with ixazomib and ixazomib citrate at 3 dpi and 5 dpi. The histological analysis of joint footpad samples were not provided due to the lack of tissue samples for this assay. And, according to the literatures [1-3], the footpad swelling and the inflammatory factors expression were the key biomarkers to assess the inflammation induced by CHIKV in mice.
Point 2: Whether the drug inhibits the virus due to its the 20S proteasome β5 inhibitors status needs to be verified experimentally.
Response 2: We agree with the comment of the reviewer. Ixazomib citrate is an orally bioavailable inhibitor of the 20S proteasome. Ixazomib refers to the biologically active boronic acid form of ixazomib citrate. Studies show that ixazomib is more selective for the β5 protease, and exerts less inhibitory activity against other active subunits. Meanwhile, ixazomib has been reported to be involved in multiple pathways[4-6], including TRAIL/death receptor signaling, β-catenin/TCF signaling and NF-κB signaling, et al. In our study, ixazomib acted at multiple stages of viruses life cycles. Therefore, the mechanisms that ixazomib inhibits TBEV, WNV, YFV and CHIKV are complex, which will be investigated in our further study. We discussed this concern in the discussion section in the revised manuscript on page 10, lines 955-972.
References:
- Gardner J, Anraku I, Le TT, Larcher T, Major L, Roques P, et al. Chikungunya virus arthritis in adult wild-type mice. Journal of virology 2010, 84(16): 8021-8032.
- Pal P, Dowd KA, Brien JD, Edeling MA, Gorlatov S, Johnson S, et al. Development of a highly protective combination monoclonal antibody therapy against Chikungunya virus. PLoS pathogens 2013, 9(4): e1003312.
- Selvarajah S, Sexton NR, Kahle KM, Fong RH, Mattia KA, Gardner J, et al. A neutralizing monoclonal antibody targeting the acid-sensitive region in chikungunya virus E2 protects from disease. PLoS neglected tropical diseases 2013, 7(9): e2423.
- Yue D, Sun X. Ixazomib promotes CHOP-dependent DR5 induction and apoptosis in colorectal cancer cells. Cancer biology & therapy 2019, 20(3): 284-294.
- Yang Y, Lei H, Qiang YW, Wang B. Ixazomib enhances parathyroid hormone-induced β-catenin/T-cell factor signaling by dissociating β-catenin from the parathyroid hormone receptor. Molecular biology of the cell 2017, 28(13): 1792-1803.
- Ganguly S, Kuravi S, Alleboina S, Mudduluru G, Jensen RA. Targeted Therapy for EBV-Associated B-cell Neoplasms. Molecular Cancer Research 2019, 17(4): 839-844.
Reviewer 2 Report
In this manuscript, Ding et al., identified ixazomib and ixazomib citrate as potential antiviral against different arboviruses. The compounds inhibited the selected arboviruses in low micromolar range and showed some in vivo efficacy in CHIKV mouse model. the results are interesting but the quality of data could be more significant by adding more readouts or experiments.
Comments:
1. did you test the activity of these compounds in other relevant cell lines to exclude that the effect is cell-dependent?
2. For the in vivo study: it is not clear on which basis you selected the doses for treatment with each compound?
3. You mentioned significant improvement of foot swelling by the compounds but there is no statistical analysis mentioned either in the text or in Fig 5B.
4. in Fig 5 C and D, it will be nice if the real viral loads were shown not only the relative amount.
5. did you quantify the viral loads in other organs of the mice to check for effect of dissemination?
6. These compounds are anticancer drugs so what is the chance that these compounds will have a clinical value to treat such viral infections i.e. benefit/risk ratio. This should be discussed in the discussion section.
Author Response
Response to Reviewer 2 Comments
Point 1: Did you test the activity of these compounds in other relevant cell lines to exclude that the effect is cell-dependent?
Response 1: We are grateful for the suggestion. We also tested the antiviral activity of those compounds in monkey kidney epithelial cells, Vero cells, which are susceptible to TBEV, WNV, YFV, and CHIKV infections. And, the 60 identified drugs all showed excellent antiviral activity at a final concentration of 5μM. And, we added a relative description in the revised manuscript on page 4, lines 462-466.
Point 2: For the in vivo study: it is not clear on which basis you selected the doses for treatment with each compound?
Response 2: We appreciate the reviewer’s attention. The dosages of ixazomib and ixazomib citrate used in the mice model referred to relative studies[1, 2] that ixazomib and ixazomib citrate were used to treat other diseases in the mice model. And the references have been added on page 3, line 311.
Point 3: You mentioned significant improvement of foot swelling by the compounds but there is no statistical analysis mentioned either in the text or in Fig 5B.
Response 3: Thanks for the kind reminder of the reviewer. Statistical analysis has been added in Fig 5B on page 8.
Point 4: In Fig 5 C and D, it will be nice if the real viral loads were shown not only the relative amount.
Response 4: Thank the reviewer’s insightful suggestion. We replaced Fig 5 C and 5D with Fig 5C in the revised manuscript on page 8, in which the absolute quantifications of viral loads at 3 dpi and 5 dpi were analyzed.
Point 5: Did you quantify the viral loads in other organs of the mice to check for effect of dissemination?
Response 5: We appreciate the reviewer’s attention. In this study, we only quantified the viral loads in the hamstring muscle tissues of the mice. According to the literature[3] and our previous experiments, in the WT C57BL/6 mice infected with the CHIKV LR2006 strain, CHIKV could be detected in muscle, spleen, lymph nodes, and liver tissues. However, the viral titer in muscle was highest compared with that of other organs. Therefore, the hamstring muscle tissues were chosen to detect the viral load in this study.
Point 6: These compounds are anticancer drugs so what is the chance that these compounds will have a clinical value to treat such viral infections i.e. benefit/risk ratio. This should be discussed in the discussion section.
Response 6: We are grateful for the valuable suggestion. The clinical value was discussed in the discussion section in the revised manuscript on page 10, lines 980-984: “As an FDA-approved drug for the treatment of multiple myeloma, the toxicity of ixazomib has been systematically evaluated [44]. While, as an anticancer drug, the toxicity is still a concern of ixazomib to be used for anti-virus infection in clinical, finding a safe therapeutic window may be feasible [45]. Before ixazomib is applied as an antiviral agent in clinical settings, much more work is still required.”.
References:
1. Chauhan D, Tian Z, Zhou B, Kuhn D, Orlowski R, Raje N, et al. In vitro and in vivo selective antitumor activity of a novel orally bioavailable proteasome inhibitor MLN9708 against multiple myeloma cells. Clinical cancer research : an official journal of the American Association for Cancer Research 2011, 17(16): 5311-5321.
2. Kupperman E, Lee EC, Cao Y, Bannerman B, Fitzgerald M, Berger A, et al. Evaluation of the proteasome inhibitor MLN9708 in preclinical models of human cancer. Cancer research 2010, 70(5): 1970-1980.
3. Gardner J, Anraku I, Le TT, Larcher T, Major L, Roques P, et al. Chikungunya virus arthritis in adult wild-type mice. Journal of virology 2010, 84(16): 8021-8032.
Round 2
Reviewer 2 Report
In their revised manuscript Ding et al, have properly addressed my previous concerns and comments, with all the correction, addition and clarification, and especially the inclusion of the suggested text, the manuscript is now ready for publication without further comments.